# Association of HSV-1 and Reduced Oral Bacteriota Diversity with Chemotherapy-Induced Oral Mucositis in Patients Undergoing Autologous Hematopoietic Stem Cell Transplantation

**DOI:** 10.3390/jcm9041090

**Published:** 2020-04-11

**Authors:** Ahreum Lee, Junshik Hong, Dong-Yeop Shin, Youngil Koh, Sung-Soo Yoon, Pil-Jong Kim, Hong-Gee Kim, Inho Kim, Hee-Kyung Park, Youngnim Choi

**Affiliations:** 1Department of Immunology and Molecular Microbiology, School of Dentistry and Dental Research Institute, Seoul National University, 101 Daehak-ro, Seoul 03080, Korea; reum7065@snu.ac.kr; 2Department of Internal Medicine, Seoul National University Hospital, 101 Daehak-ro, Seoul 03080, Korea; hongjblood@snu.ac.kr (J.H.); stephano.dyshin@gmail.com (D.-Y.S.); go01@snu.ac.kr (Y.K.); ssysmc@snu.ac.kr (S.-S.Y.); 3Cancer Research Institute, Seoul National University, 101 Daehak-ro, Seoul 03080, Korea; 4Biomedical Knowledge Engineering Laboratory, School of Dentistry, Seoul National University, 1 Kwanak-ro, Seoul 08826, Korea; childfeel@naver.com (P.-J.K.); hgkim@snu.ac.kr (H.-G.K.); 5Department of Oral Medicine and Oral Diagnosis, School of Dentistry and Dental Research Institute, Seoul National University, 101 Daehak-ro, Seoul 03080, Korea

**Keywords:** oral mucositis, risk factor(s), microbiota, HSV-1

## Abstract

Oral mucositis (OM) is a common complication of chemotherapy and remains a significant unmet need. The aim of this study was to investigate the role of oral bacteriota and HSV-1 in OM. Forty-six patients admitted for autologous hematopoietic stem cell transplantation were longitudinally evaluated for OM, Candida, HSV-1, and leukocyte count, and buccal mucosal bacterial samples were obtained during their admission period. The bacterial communities collected at the baseline and post-chemotherapy, chosen from the time with the highest severity, were analyzed by sequencing the 16S rRNA gene. Twenty (43.5%) patients developed OM, the severity of which ranged from 1 to 5 according to the Oral Mucositis Assessment Scale (OMAS). Chemotherapy significantly increased the prevalence of HSV-1 detection but not that of Candida. The bacterial communities of patients after conditioning chemotherapy were characterized by aberrant enrichment of minor species and decreased evenness and Shannon diversity. After adjustment for age, gender, and neutropenia, the presence of HSV-1 was associated with the incidence of OM (odds ratio = 3.668, *p* = 0.004), while the decrease in Shannon diversity was associated with the severity of OM (*β* = 0.533 ± 0.220, *p* = 0.015). The control of HSV-1 and restoration of oral bacterial diversity may be a novel option to treat or prevent OM.

## 1. Introduction

Oral mucositis (OM) refers to erythematous and/or ulcerative lesions in the oral cavity that occur secondary to cancer therapy. It is a common complication of chemotherapy. Patients who receive a concomitant radiation therapy in the head and neck area, intensive chemotherapy for acute leukemia, or myeloablative conditioning for hematopoietic stem cell transplantation (HSCT) are at particularly high risk for OM [1]. The prevalence of OM in HSCT patients has been reported to be 49–99% [2,3,4]. OM is associated with severe pain, increased use of opioid analgesics, dysphagia, interruption of planned chemotherapy, bloodstream infection, and increased medical costs [1]. Although clinical practice guidelines for the intervention and management of OM have been published by the Multinational Association of Supportive Care in Cancer, OM remains a significant unmet need [5]. A clear understanding of the etiopathological mechanisms for OM is important to the development of new, effective, targeted therapies.

According to current knowledge, OM is initiated by the generation of reactive oxygen species and the release of damage-associated molecular pattern molecules in response to cytotoxic reagents. Subsequent signaling through innate immune receptors results in NF-κB-dependent upregulation of inflammatory cytokines such as TNF*α*, IL-1, and IL-6. TNFα also activates NF-κB, leading to a positive feedback and signal amplification. Ultimately, apoptosis of epithelial stem cells leads to ulceration [6]. Therefore, the primary risk factor for the incidence and severity of OM is the dose and type of chemotherapy [1]. Microbial-associated molecular pattern molecules can also activate the innate immune responses and NF-κB. Thus, the roles of oral viruses, fungi, and bacteria, particularly herpes simplex virus (HSV), *Candida albicans*, and periodontopathogens, in the etiopathogenesis of OM have been considered for years, but some conflicting data have been reported [7,8,9,10].

We previously reported that OM has a strong association with HSV-1 but no association with Candida, plaque index, or decayed, missing, and filled surface (DMFS) scores in patients with hematological malignancies undergoing intensive chemotherapy or HSCT [2]. However, only 46% of patients with positive HSV-1 detection developed OM, suggesting the role of additional factors. In this study, we aimed to investigate the role of oral bacteriota and HSV-1 in the incidence and severity of OM. To reduce the number of variables that can affect oral bacteriota, the current study restricted subjects to patients undergoing autologous HSCT.

## 2. Experimental Section

### 2.1. Study Design and Sampling

This study was carried out in accordance with the Declaration of Helsinki after approval from the Institutional Review Board at Seoul National University School of Dentistry (S-D20160016) and conforms to the STROBE guidelines. Written informed consent was obtained from all participants.

The study design is summarized in Figure 1A. The inclusion criteria for the patient group were as follows: (1) age ≥ 19 years and (2) admitted for autologous HSCT to Seoul National University Hospital. The exclusion criteria were as follows: (1) any prior radiation therapy or surgery within the 3 weeks prior to enrollment; (2) patients with definitive symptoms or signs of OM at baseline; (3) those with other severe dental diseases or systemic diseases that could significantly affect the study outcomes, including active infection, uncontrolled diabetes, and symptomatic cardio- or cerebrovascular disorders; (4) smokers; and (5) those who had underlying psychological diseases or cognitive disorders that precluded necessary communication. Twenty-two patients were included in our previous study [2]. For the control group, the inclusion criterion was age ≥ 19 years, and the exclusion criteria were as follows: (1) smokers; (2) the use of antibiotics or steroid within the last month; (3) any medication for systemic diseases; and (4) subjects with any lesions in the oral mucosa. Sample size was estimated base on the prevalence of OM and the structure of mucosal bacteriota in previous studies [2,11]. All of the enrolled patients received a basic dental examination that included panoramic radiography and visual examination for the presence of potential foci of infection, oral mucosal diseases, and any abnormal changes in the oral mucosa. Evaluations of OM, HSV-1, and *Candida* spp., and sampling of oral bacteria were performed at baseline (day 0 ± 2), week 2 (day 14 ± 2), week 3 (day 21 ± 2), and week 4 (day 28 ± 2), while the patients were hospitalized. In our study, day 0 was defined as the day of initiation of conditioning chemotherapy. Except for the oral examination and sampling, all of the other procedures, including oral care, were not changed from the institutional protocols.

OM was graded according to the National Cancer Institute Common Toxicity Criteria for Adverse Events (NCI-CTCAE) v3.0 [12] and Oral Mucositis Assessment Scale (OMAS) [13]. Among the oral lesions, typical HSV-associated vesicular lesions and ulcers located on non-movable keratinized mucosa observed in the patient with HSV-1 detection were not considered OM.

The buccal mucosa and dorsum of the tongue were swabbed with sterile cottons to evaluate *Candida* spp. A sterilized 30 × 30 mm Immobilon-P Transfer Membrane (Merck Millipore, Billerica, MA, USA) was placed on the buccal mucosa for 30 s and was subjected to DNA extraction to evaluate HSV-1 and bacteriota. When OM developed on the buccal mucosa, the sampling site included the lesions. Oral samplings were performed between meals.

Control subjects were asked to avoid eating and tooth brushing for two hours before the visit. They received visual examination for the presence of oral mucosal diseases and any abnormal changes in the oral mucosa, and the buccal mucosal sample was obtained using the membrane.

### 2.2. Autologous HSCT Procedures

For all of the patients, prophylactic antimicrobial and antifungal drugs were administered from day 0 to the date of neutrophil engraftment, defined as > 1000/μL of the absolute neutrophil count (ANC) for 3 consecutive days, according to the Center for International Blood and Marrow Transplant Research guidelines [14]: ciprofloxacin 500 mg orally twice a day and micafungin 50 mg via intravenous infusion once a day were prescribed for gut decontamination and prophylaxis against invasive fungal infections, respectively. Two patients with multiple myeloma had received anti-cancer therapy that is associated with a high risk of developing herpes zoster; thus, were under the prophylactic acyclovir before the start and during the entire study period. Ten patients received acyclovir for 3–15 days during the study period to treat herpes zoster, disseminated zoster, or severe oral ulceration.

Several regimens of myeloablative high-dose chemotherapy were used as conditioning therapy before stem cell infusion. Body irradiation was not included in the conditioning therapy (Appendix A). Previously collected and cryopreserved autologous CD34^+^ HSCs were infused according to the clinical protocols. Patients were discharged after successful neutrophil engraftment and ≥ 4 days of platelet transfusion independency without major acute post-HSCT complications.

During hospitalization, i.e., the entire study period, patients rinsed their oral cavity with normal saline 4 times daily, with chlorhexidine twice daily, and with 5–10 mL of nystatin oral suspension 3 times daily immediately after a meal to reduce the risk of oral infection, including oropharyngeal candidiasis. Standard infection prevention measures were applied according to the institution’s policy, including isolation in a laminar flow room with a high-efficiency particulate air-filter, hand washing and wearing masks when contacting personnel, and a low bacterial diet until neutrophil engraftment.

### 2.3. Evaluation of Candida spp. and HSV in the Oral Cavity

The presence of *Candida* spp. was determined by plating cotton swabs onto ChromID^®^ Candida Agar (BioMérieux, Lyon, France) and by incubation at 37 ℃ under aerobic conditions for 48 h. The recorded *Candida* spp. included *C. albicans*, which forms dark blue colonies, *C. tropicalis*, *C. lusitaniae*, and *C. kefy*, which form pink colonies, and *C. dubliniensis*, which forms turquoise or white colonies, on the ChromID Candida agar. However, the results were presented as *Candida spp*. Genomic DNA was extracted from the membrane using the DNeasy^®^ Power soil kit (Qiagen, Hilden, Germany) and eluded in 100 μL water. The presence of the HSV genome in the oral mucosa was determined by PCR using the isolated DNA and the HSV 1/2 PCR kit (BioCore, Seoul, South Korea) as previously described [2].

### 2.4. Analysis of Bacterial Communities

Bacterial 16S rRNA gene fragments spanning the hypervariable region V3–V4 were amplified by PCR using the primers 341F_805R. The amplified products were sequenced using the Illumina Miseq system (San Diego, CA, USA) performed at ChunLab Inc. (Seoul, Korea). All bioinformatic analyses were performed using the integrated database EzBioCloud (http://www.ezbiocloud.net/) and CLcommunity™ software ver3.46 provided by ChunLab as previously described [15,16]. Uncharacterized taxa mentioned in this article were searched against the Human Oral Microbiome Database (http://www.homd.org/), and oral taxon numbers were assigned when available. The sequence data are available in the NCBI Sequence Read Archive under BioProject accession PRJNA580493.

### 2.5. Statistical Analysis

The baseline data between groups were compared by t-test for age and either chi-square or Fisher’s exact test for categorical data. Detection incidences of HSV-1 and *Candida* spp. at the baseline and post-chemotherapy were compared by the McNemar’s test. The associations of microbial factors with OM after chemotherapy were determined by generalized estimating equations for analysis of repeated measures with adjustment for age, gender, and severe neutropenia that were identified as confounding factors from the analysis of 125 evaluations, including the baseline data. To determine differences in the alpha-diversity indexes and relative abundance of microbial taxa between groups, either a parametric (t-test or ANOVA) or non-parametric approach (Mann–Whitney or Kruskal–Wallis) was used depending on whether the datasets had normal distributions. A paired t-test or Wilcoxon signed-rank test was used to compare baseline and post-chemotherapy samples from the same subjects. Linear discriminant analysis (LDA) effect size (LEfSe) analysis of oral bacteriota was performed online (https://huttenhower.sph.harvard.edu/galaxy/) with setting thresholds on the logarithmic LDA score > 2.0 and *p* < 0.05. The correlation between the relative abundance of microbial taxa and either the OMAS or the Shannon index was determined by the Spearman’s rank test. Differences in the clustering of bacterial communities between groups were determined by the permutational multivariate analysis of variance (PERMANOVA) test using Adonis function of the R vegan package (R Foundation for Statistical Computing, Vienna, Austria). The significance threshold was adjusted using the Benjamini–Hochberg false discovery rate method in R. Other tests were performed by SPSS 23.0 (IBM, Armonk, NY, USA). A case with missing data was omitted from the analysis of the missing variable. The data were considered statistically significant at a *p* value < 0.05.

## 3. Results

### 3.1. OM Is Associated with the Detection of HSV-1

From July 2016 to June 2018, we recruited 49 patients admitted for autologous HSCT and followed them at baseline, week 2, week 3, and week 4 from the start of chemotherapy as long as they were hospitalized. Three patients were excluded due to diverse reasons. Because the hospitalization periods varied, 125 evaluations of 46 patients were performed (Figure 1A). Twenty (43.5%) patients developed OM defined as the NCI-CTCAE grade > 0, which was first observed at week 2 or 3 and was observed more than twice in four patients (Figure 1B). The highest scores for OM ranged from 1 to 3 on the NCI-CTCAE scale and from 1 to 5 on the OMAS (Figure 1C). Among the demographic, clinical, and microbial parameters at the baseline, none were associated with the incidence of OM (Table 1). Chemotherapy significantly increased the prevalence of HSV-1 detection (*p* = 0.013) but not that of *Candida* spp. (*p* = 0.338; Figure 1D). The increase in HSV-1 detection after chemotherapy was particularly evident in males (Figure 1E). Chemotherapy also induced significant decreases in the blood neutrophil and lymphocyte counts at week 2 (Figure 1F). In the analysis of the 79 evaluations performed after chemotherapy, *Candida* detection was not associated with OM, but HSV-1 detection increased the risk of developing OM by 3.7-fold after adjustment for age, gender, and neutropenia (Table 2). However, 11 of 27 patients with post-chemotherapy HSV-1 positivity did not develop OM, and four patients experienced OM in the absence of HSV-1, suggesting involvement of other factors. 

### 3.2. Chemotherapy-Accompanied Dysbiosis of Buccal Mucosal Bacteriota Is Associated with Severe OM

To understand the role of oral bacteria in OM, buccal mucosal samples collected at the baseline and post-chemotherapy were analyzed by sequencing the 16 rRNA gene. The post-chemotherapy samples were chosen from the time with the highest NCI-CTCAE score for the patients with OM (week 2: *n* = 15 (75%), week 3: *n* = 4 (20%), week 4: *n* = 1 (5%)) and at similar frequencies for the patients without OM (week 2: *n* = 20 (77%), week 3: *n* = 6 (23%)). Control samples (*n* = 15) obtained from subjects without oral mucosal diseases were also analyzed. 

Compared with the control communities, the baseline communities of patients had comparable alpha diversities. Chemotherapy significantly decreased the Shannon diversity, an index that accounts for both richness and evenness of the species present. The decreased Shannon diversity was attributed to the decreases in both species richness and evenness determined by the Chao1 and Simpson indexes, respectively. UniFrac-based principal coordinate analysis (PCoA) and PERMANOVA tests revealed a significant shift in the clustering of the post-chemotherapy communities from the baseline communities of patients (Figure 2A). 

We explored whether the experience of OM was associated with differences in the structure of oral bacteriota at the baseline or after chemotherapy. Different from our expectation, no differences were observed between the OM-Free and OM-Experienced groups at both time points (Figure 2B). When the post-chemotherapy communities were divided by OM severity, however, the group with an NCI-CTCAE score ≥ 2, i.e., ulceration, presented a reduced Shannon diversity, which was attributed to the decrease in species evenness. After adjustment for age, gender, and neutropenia, the decrease in the Shannon index by 1 increased the OMAS score by 0.533 (Table 2). A PCoA plot also revealed a significant shift in the clustering of communities with NCI-CTCAE scale ≥ 2 from the other communities (Figure 2C). These differences were not observed in the baseline samples. Collectively, the chemotherapy-accompanied decreases in mucosal bacterial diversity was associated with severe OM with ulceration.

### 3.3. The Presence of HSV-1 after Chemotherapy Is Associated with the Dysbiosis of Buccal Bacteriota

We next explored whether the presence of *Candida* spp. or HSV-1 affected the structure of oral bacteriota. Neither the alpha nor beta diversities of oral microbiota differed by the presence of *Candida* spp. both at the baseline and after chemotherapy (Appendix A). Interestingly, the presence of HSV-1 was associated with reduced species evenness and Shannon diversity only in the post-chemotherapy communities, but no significant differences were found in the clustering of four groups (Figure 2D). When the post-chemotherapy communities were divided into four groups by the presence of HSV-1 and OM, no differences were noted in either the alpha diversities or clustering by the presence of OM among the HSV-1-negative and HSV-1-positive communities (Appendix A). Collectively, the presence of HSV-1 aggravated the chemotherapy-accompanied decrease in mucosal bacterial diversity.

### 3.4. The Severity of OM Is Correlated with Depletion of Commensals

We explored whether changes in specific taxa were associated with chemotherapy-accompanied dysbiosis, OM development, or the presence of either *Candida* spp. or HSV-1. At the phylum level, significant differences were observed only between the baseline and post-chemotherapy communities of patients: Firmicutes and Tenericutes were enriched at the expense of Bacteroidetes and Fusobacteria after chemotherapy (Figure 3A). At the species level, only 15 species were differentially distributed between the control and baseline communities of patients (Appendix A). However, the overview of species composition revealed aberrant enrichment of minor species in the patients, which was more evident after chemotherapy than at the baseline. A single species that normally accounted for less than 1% of a community in controls often expanded to greater than 20% and sometimes even greater than 80% in the patients (Figure 3B). Because the enriched species varied depending on the subject, 90 of 92 species that significantly changed after chemotherapy showed decreases in relative abundance (Appendix A). The presence of HSV-1 was associated with several bacterial taxa, and the associated taxa at the baseline were different from those found after chemotherapy (Figure 4). No differences were observed in the relative abundance of specific species by OM experience, OM severity, or the presence of *Candida* spp.

Potential associations between OM severity and specific species were further analyzed by Spearman’s rank correlation analysis between OMAS scores and the relative abundance of each taxa. OM severity was negatively correlated with the relative abundance of 27 species, some from the health-associated genera *Streptococcus*, *Rothia*, *Actinomyces*, *Gemella*, *Veillonella*, and *Stomatobaculum* and others from the genera *Leptotrichia* and *Prevotella*. Twenty-six of 27 species also had significant positive correlations with the Shannon index (Figure 5). No species had a positive correlation with OMAS scores. Collectively, OM severity was correlated with the depletion of species that may maintain complex health-associated commensal communities on the mucosal surface.

## 4. Discussion

The microbiota colonizing the oral mucosa contributes to normal mucosal physiology and maintains homeostasis with hosts in health. Chemotherapy-induced myelosuppression is known to be followed by microbial dysbiosis, and the role of oral microbiota in the etiopathogenesis of OM has been suggested for years [17]. In this study, we demonstrate that chemotherapy-accompanied activation of HSV-1 and dysbiosis of oral mucosal bacteriota are associated with the incidence and severity of OM, respectively, in patients undergoing autologous HSCT. 

Because HSCT involves profound immunosuppression, all measures for infection prevention and control were maintained throughout the study. The variable use of acyclovir during the study period may have had confounding effects on HSV-1 detection and development of OM. Whereas the 2 patients under the prophylactic acyclovir experienced neither HSV-1 activation nor OM throughout the study period, 7 and 6 of 10 patients who received acyclovir therapy for a short period had post-chemotherapy HSV-1 positivity and OM, respectively. Combining the prophylactic and therapeutic purposes, the use of acyclovir did not affect either the post-chemotherapy HSV-1 positivity or the incidence of OM in the current study (Appendix A), probably because the number of prophylactic cases was small and the therapeutic acyclovir was administered after detection of HSV-1 in the oral cavity in most cases. Importantly, HSV-1 was detected in less than 5% of samples and was not associated with OM in a study where all patients were subjected to anti-herpes prophylaxis during allogeneic HSCT procedure [18]. Despite the uniform prophylactic administration of micafungin and the use of nystatin gargle, colonization with *Candida* spp. was observed in 18 patients, which may be attributed to anti-fungal resistance in *Candida* [19]. All patients were also under prophylactic ciprofloxacin and used chlorhexidine gargle during the study period. Notably, the baseline bacterial communities of 20 patients were exposed to ciprofloxacin and chlorhexidine for ≤ 4 days, which should be considered in interpretation of the data that compared the baseline bacterial communities of patients with the control communities.

Post-chemotherapy bacterial communities were characterized by decreased Shannon diversity attributed to decreases in both species richness and evenness. The decrease in the species evenness was explained by the substantial enrichment of one or two species, likely leading to significant decreases in the species richness and relative abundance of 91 species. Two groups have reported different results for the changes in Shannon diversity of oral bacteriota after chemotherapy in patients with solid tumors: no change in mucosal microbiota [11] and a decrease in salivary microbiota but a slight increase in mucosal microbiota [20]. The reasons for the discrepancies among the three studies including ours are not clear because differences exist in many variables such as age, tumor type, chemotherapy type, antibiotic use, and oral care protocols. Unique characteristics of our cohort, including uniform antibiotic prophylaxis, may have contributed to the reduced Shannon diversity after chemotherapy. The enrichment of Firmicutes at the expense of Bacteroidetes and Fusobacteria after chemotherapy observed in this study could be caused by ciprofloxacin, which is more effective against Gram-negative than against Gram-positive bacteria [21]. It has been reported that administration of systemic antibiotics is associated with the diversity loss of the gut microbiome in patients receiving allogeneic HSCT [22]. A very recent study by Laheji et al. also reported decreased oral bacterial diversity after chemotherapy in patients who were undergoing autologous HSCT and received prophylactic ciprofloxacin [23]. The role of chlorhexidine cannot be excluded, although its effect on the structure of oral microbiota remains to be clarified. 

Knowledge of inter-kingdom interactions in the oral microbiome is limited. While the presence of *Candida* spp. did not affect the structure of oral bacteriota, the presence of HSV-1 was interestingly associated with decreased species evenness and Shannon diversity only after chemotherapy. In addition, different taxa from the baseline and post-chemotherapy bacterial communities were associated with the presence of HSV-1. Potential interaction among HSV-1, oral bacteriota, and host neutrophils needs to be further studied.

While the presence of HSV-1 had an association with the incidence of OM, the decrease in Shannon diversity was associated with the severity of OM. Therefore, HSV-1 may first trigger the development of OM, and dysbiotic oral bacteriota may aggravate the lesion. The association of decreased bacterial diversity with severe OM with ulceration coincides with the finding from the previous study by Laheji et al. [23].

Although the seropositivity of HSV-1 was not tested in the current study, primary HSV-1 infection manifestation was not observed in any patient. Considering the 100% HSV-1 seropositive rate in Koreans aged ≥ 30 years [24], HSV-1 detected in the oral mucosa seems to be the result of recurrence. The high prevalence of HSV-1 detection in patients at baseline but none in control subjects must be attributed to the lymphocytopenic condition of patients because the maintenance of latency primarily depends on noncytolytic CD8^+^ T cells [25]. Contribution of myelosuppressive therapy to HSV-1 activation has also been reported [26]. Only five of 49 observations with HSV-1 detection accompanied viral lesions. Asymptomatic viral shedding of HSV-1 in seropositive subjects without a history of recurrent herpes labialis has been reported, in which HSV-1 DNA was detected in the buccal mucosa, lower lip, and dorsum of the tongue at a similar frequency [27]. HSV-1 activates TLR2 and TLR9 and induces the production of inflammatory cytokines IL-1 and IL-6 and chemokines CCL3 and CCL4 from keratinocytes [28,29]. Thus, together with the reactive oxygen species and inflammatory cytokines induced by the cytotoxic agents, HSV-1 may contribute to the development of OM.

How oral bacterial dysbiosis aggravates OM severity is not clear from the current study. Associations of *Porphyromonas gingivalis* and *Fusobacterium nucleatum* with oral ulceration were reported in a study where seven periodontopathic bacteria were tested by PCR in patients undergoing allogeneic HSCT [9]. Laheji et al. [23] reported associations of *Staphylococcus* spp. and *Enterococcus* spp. with ulcerative OM during autologous HSCT. Hong et al. [20] reported correlations of OM severity with the enrichment of *F. nucleatum*, *Treponema maltophilum*, *Clostridiales* HOT093, and *Prevotella oris*, but the depletion of 29 species in patients who received 5-fluorouracil- or doxorubicin-based chemotherapy. They also demonstrated the pro-inflammatory and pro-apoptotic capacity of *F. nucleatum* but toleration of mucositis-depleted *Streptococcus salivarius* by cultured oral keratinocytes. In our study, aberrant enrichment of *Staphylococcus epidermidis*, *Enterococcus durans*, or *E. faecalis* was observed in 7 patients. Five of the 7 patients had ulcerative OM, which partly agrees with the finding by Laheji et al. [23]. However, enriched species varied from patient to patient, and no taxa showed a positive correlation with OMAS scores. In contrast, 27 species were negatively correlated with the OMAS scores. Among them, *Streptococcus parasanguinis*, *Veillonella atypica*, *Actinomyces* sp. HOT172, *Prevotella pallens*, *Oribacterium sinus*, and *Lachnoanaerobaculum orale* were also reported by Hong et al. [20]. Although most of the negatively correlated species are members of commensals, several *Prevotella* and *Leptotrichia* species were also included as reported by Hong et al. [19]. The *Prevotella* and *Leptotrichia* species showed moderate positive correlations with the Shannon diversity. Thus, they may contribute to the formation of complex bacterial communities through interaction with diverse species, as they do in dental plaque [30]. Ultimately, the host-microbe interactions of health-associated communities that benefit normal mucosal physiology but are tolerated by immune systems need to be clarified.

The sampling sites of eight post-chemotherapy samples included lesions, two of which had ulceration. When those eight samples were segregated from the samples obtained from the healthy sites of patients with OM, no inter-group difference was observed in any parameter of the bacterial communities. This finding coincides with a previous report that mucosal bacteriota collected from the healthy, erythematous, and ulcerated sites of the same subjects with chemotherapy-induced OM were not different [20].

This study has a number of limitations. First, ciprofloxacin, chlorhexidine, micafungin, and nystatin were uniformly used, and their role in the chemotherapy-accompanied dysbiosis of buccal mucosal bacteriota cannot be clarified. Second, a relatively small sample size limited the statistical power when the subjects were subdivided by the presence of HSV-1 and OM experience. Third, one of our original aims to find an additional factor other than HSV-1 involved in the incidence of OM was not achieved.

In conclusion, the chemotherapy-accompanied activation of HSV and decrease in the bacterial diversity were associated with the incidence and severity of OM, respectively. A clinical trial to evaluate the effect of prophylactic anti-herpes drugs on the prevention of OM in patients undergoing autologous HSCT is under way. Although the use of prophylactic antibiotics during the neutropenic period in HSCT has been a standard protocol for decades, reevaluation of this practice has been recently proposed due to rising rates of antibiotic resistance and adverse effects of perturbed microbiome [31]. The adverse effects of perturbed oral microbiome may include aggravation of OM. Likewise, the use of chlorhexidine gargle for oral care also needs to be reevaluated. In addition, restoration of microbial diversity using salivary microbiota may be a novel therapeutic option to prevent severe OM.

## Figures and Tables

**Figure 1 jcm-09-01090-f001:**
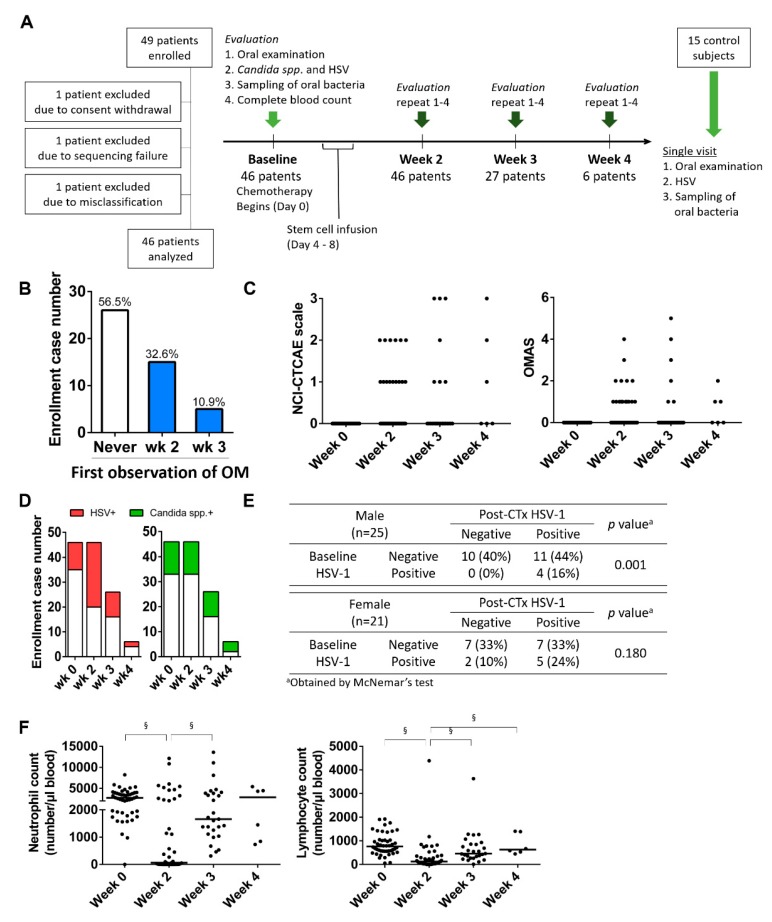
Incidence of oral mucositis (OM) and changes in the detection of HSV-1 and *Candida* spp. in the oral cavity and blood leukocyte counts during autologous hematopoietic stem cell transplantation (HSCT). (**A**) Flow chart of study. (**B**) Incidence of OM defined as the National Cancer Institute Common Toxicity Criteria for Adverse Events (NCI-CTCAE) grade > 0. (**C**) Severity of OM evaluated by the NCI-CTCAE scale and Oral Mucositis Assessment Scale (OMAS). (**D**) Detection incidence of HSV-1 and *Candida* spp. in the oral cavity. (**E**) Detection incidence of HSV-1 in different genders. (**F**) Changes in the blood neutrophil and lymphocyte counts. §, *p* < 0.05 by the Kruskal–Wallis test followed by post-hoc with the Bonferroni method.

**Figure 2 jcm-09-01090-f002:**
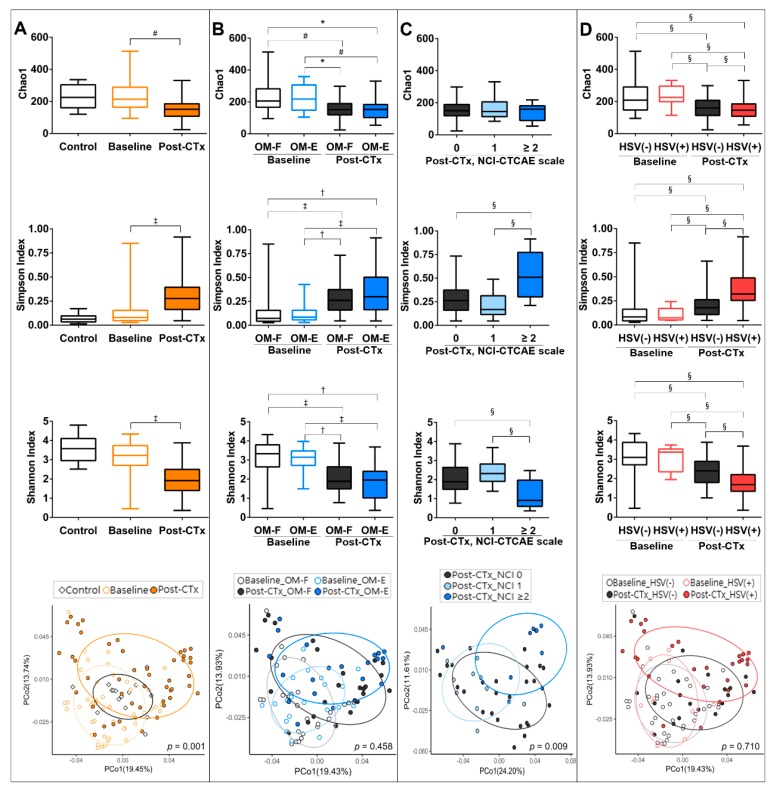
Structures of the mucosal bacterial communities collected from control subjects and patients undergoing autologous HSCT at baseline and post-chemotherapy (post-CTx). Bacterial communities were analyzed by high-throughput sequencing of the bacterial 16S rRNA gene. The species richness, evenness, and diversity were estimated by the Chao1, Simpson, and Shannon indexes, respectively (Top three panels). A PCoA plot was generated using the weighted UniFrac metric with normalization for read counts (bottom-most panel). (**A**) The communities of control subjects and the baseline and post-chemotherapy communities of patients were compared. (**B**) The baseline and post-chemotherapy communities of patients who were free of OM (OM-F) and those of patients who experienced OM (OM-E) defined as the NCI-CTCAE grade > 0 during autologous HSCT were compared. (**C**) The post-chemotherapy communities of patients were compared by the NCI-CTCAE scale. (**D**) The baseline and post-chemotherapy communities of patients without or with HSV-1 detection were compared. *, *p* < 0.05 by non-paired t-test; #, *p* < 0.05 by paired t-test; †, *p* < 0.05 by the Mann–Whitney U test; ‡, *p* < 0.05 by the Wilcoxon signed-rank test; §, *p* < 0.05 by the Kruskal–Wallis test followed by post-hoc with the Bonferroni method. The *p* values in the PCoA plots were obtained by PERMANOVA.

**Figure 3 jcm-09-01090-f003:**
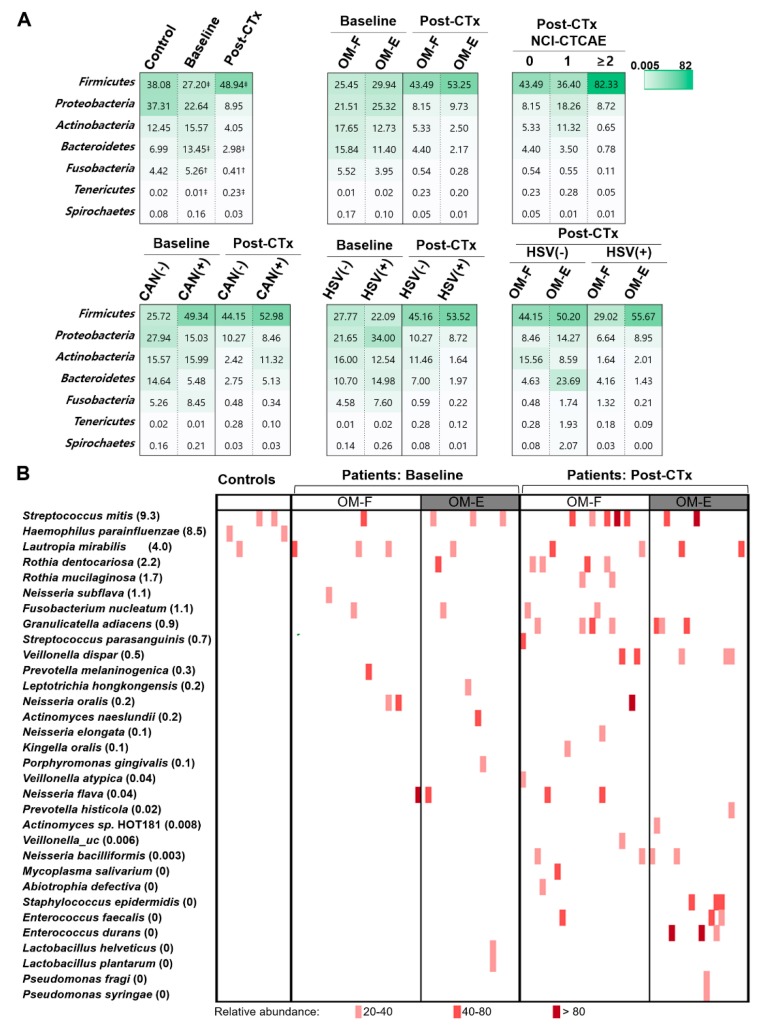
Composition of the mucosal bacterial communities collected from control subjects and patients undergoing autologous HSCT at baseline and post-chemotherapy (post-CTx). (**A**) The relative abundances of major phyla (average relative abundance > 1%) are expressed as a heat map. The value in each box presents the median. ‡ indicates a pair with a significant difference (*p* < 0.05) by the Wilcoxon rank-sum test. (**B**) Species with a relative abundance > 20% in any sample were chosen and presented as a heat map. The number inside parentheses next to the name of the species indicates a median value for the relative abundance in the control group.

**Figure 4 jcm-09-01090-f004:**
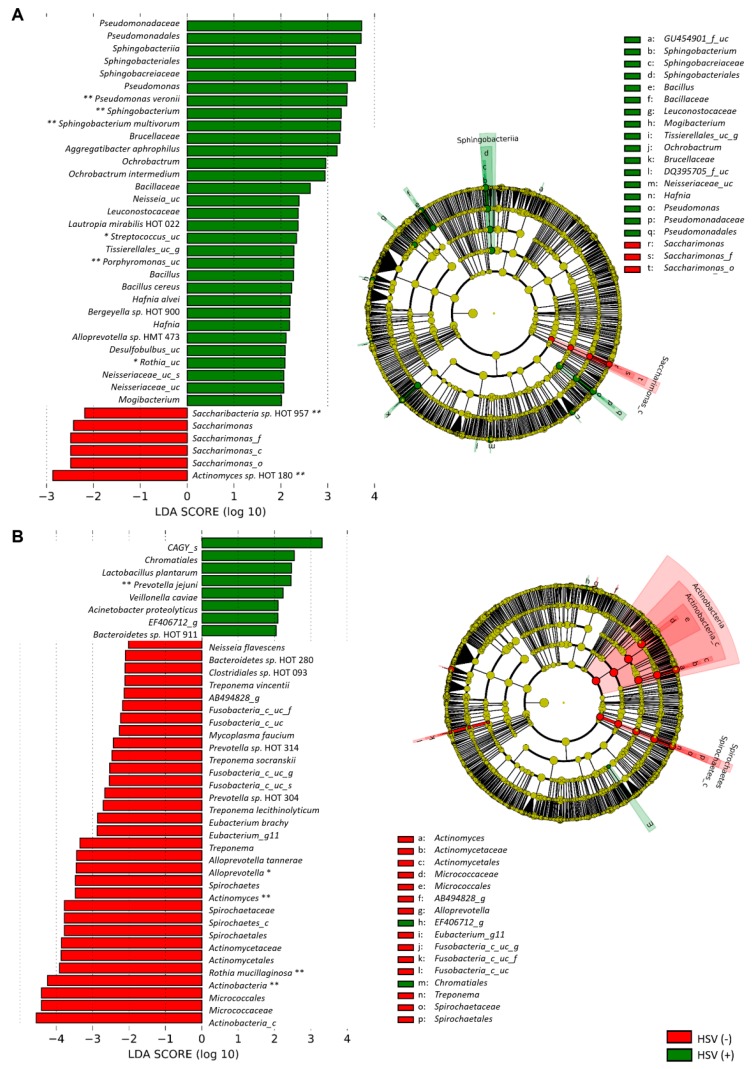
Linear discriminant analysis (LDA) effect size (LEfSe) analysis of oral bacteriota by the presence of HSV-1. Left panels show the most differentially distributed taxa with *p* < 0.05, LDA score > 2, and LEfSe cladograms (right panels) present taxonomic hierarchical trees of phylogenetic oral bacteriota distribution by the presence of HSV-1 at the baseline (A) and post-chemotherapy (B). Asterisks indicate taxa that passed the Benjamini–Hochberg false discovery test (*, *p* < 0.1; **, *p* < 0.05).

**Figure 5 jcm-09-01090-f005:**
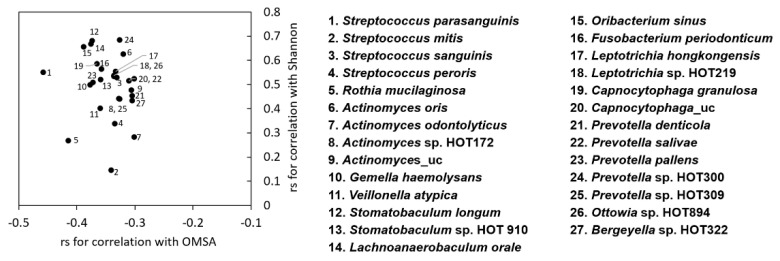
Species negatively associated with the severity of oral mucositis. Correlations of the relative abundance of each species with the OMAS score and Shannon index were analyzed by the Spearman’s rank correlation test. Twenty-seven species, the relative abundance of which showed non-negligible correlations (r_s_ ≥ │0.3│, *p* ≤ 0.004) with OMAS scores, were chosen. The graph presents the coefficients for correlation with OMAS and those for correlation with the Shannon index.

**Table 1 jcm-09-01090-t001:** Subject characteristics at baseline and their associations with oral mucositis.

	Control (*n* = 15)	Patients Undergoing Autologous HSCT			*p* Values	
		Total (*n* = 46)	OM-Free (*n* = 26)	OM-Ex (*n* = 20)	*p* ^1^	*p* ^2^
**Age**	38.1±11.6	52.7 ± 9.8	52.0 ± 10.0	53.5 ± 9.8	< 0.001 ^a^	0.626 ^a^
**Gender (% male)**	40	54.3	42.3	70	0.338 ^b^	0.062 ^b^
**Type of disease**						
Multiple myeloma (%)		60.9	65.4	55		0.714 ^b^
Lymphoma (%)		32.6	26.9	40		
Others (%)		6.5	7.7	5		
**HSV-1 (%)**	0	23.9	19.2	30	0.038 ^b^	0.401 ^b^
***Candida*** **spp. (%)**		37	34.6	40		0.711 ^b^
**ANC < 500/μL** **(%)**		2.2	0	5		0.254 ^b^
**ALC < 1000/μL** **(%)**		69.6	65.4	75		0.487 ^b^
**Prophylactic antibiotic use (% yes)**		100	100	100		0.249 ^b^
**Prophylactic antifungal use (% yes)**		100	100	100		
**Prophylactic acyclovir use (% yes)**		4.3	15.4	0		0.498 ^b^

HSCT, hematopoietic stem cell transplantation; OM, oral mucositis; HSV-1, herpes simplex virus-1; ANC, absolute neutrophil count; ALC, absolute lymphocyte count. *p ^1^*: control vs. total patients *p ^2^*: patients who did not develop OM (OM-Free) vs. those who experienced OM (OM-Ex) defined as the NCI-CTCAE grade > 0 during chemotherapy ^a^ Obtained by *t*-tests, ^b^ Obtained by chi-square or Fisher’s exact test.

**Table 2 jcm-09-01090-t002:** Associations of post-chemotherapy microbial factors with the incidence and severity of oral mucositis in patients undergoing autologous hematopoietic stem cell transplantation.

Variables		OM		OM (yes vs. no)			OM Severity (OMAS)	
		No (*n* = 54) ^a^	Yes (*n* = 25) ^a^	Odds Ratio	95% CI	*p* ^b^	*β*	*p* ^b^
HSV-1	Negative	63%	25%	3.668	1.512–8.895	0.004	0.321 ± 0.223	0.151
	Positive	37%	75%					
*Candida* spp.	Negative	64.8%	66.7%	1.008	0.362–2.807	0.988	0.064 ± 0.215	0.765
	Positive	35.2%	33.3%					
Bacterial diversity ^c^		1.89	1.95	0.604	0.275–1.328	0.210	−0.533 ± 0.220	0.015
		(0.77–3.88)	(0.36–3.68)					

OM, oral mucotitis defined as the NCI-CTCAE grade > 0; OMAS, oral mucositis assessment scale; HSV-1, herpes simplex virus-1. ^a^ Evaluation numbers for HSV-1 and *Candida spp*. *n* = 24 for OM (no sampling due to too severe OM); Evaluation numbers for the Shannon index are *n* = 26 for no OM and *n* = 20 for OM. ^b^ Obtained by generalized estimating equations for analysis of repeated measures (adjusted for age, gender, and neutropenia). ^c^ Values represent the median (range) of Shannon index.

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
