# Peer review of "Association of HSV-1 and Reduced Oral Bacteriota Diversity with Chemotherapy-Induced Oral Mucositis in Patients Undergoing Autologous Hematopoietic Stem Cell Transplantation"

_jcm, 2020, doi:10.3390/jcm9041090_

Round 1

Reviewer 1 Report

In the paper: “Association of HSV-1 and Reduced Oral Bacteriota  Diversity with Chemotherapy-Induced Oral Mucositis in Patients Undergoing Autologous  Hematopoietic Stem Cell Transplantation” the Authors reported the presence of HSV-1 as significantly associated with the incidence of Oral Mucositis while the decrease in microbial diversity assessed by Shannon index is associated with the severity of OM .

I believe that the Authors showed an interesting work and deserved to be published. 
However, the paper draft really needs major revisions. 

  1. The Authors reported that chemotherapy significantly increased the prevalence of HSV-1 detection: in this context it would be interesting to see the results divided by gender
  2. The comparison with the control group should not be the subject of statistical analysis since all patients have been treated with antibiotics
  3. if anti-fungal therapy was given during the course of the study to all patients: how can the results regarding Candida infection be valid?
  4. The Authors say that the presence of HSV1 is associated with reduced species eveness and Shannon diversity only in the post-chemotherapy community. In addition to comparing indexes of microbial diversity I strongly suggest to search significant enrichment of microbial taxa in HSV1+ patients compared to HSV1- by LEFSE analysis. Data from heatmap are not sufficient.
  5. The Authors say: Interestingly, the use of acyclovir was associated with the increased incidence of OM (Odds ratio 3.982, 95% CI 319 1.173–13.514, = 0.027), reflecting increased HSV-1 detection among the acyclovir recipients and the therapeutic use of acyclovir.The answer is: how much did the prophylactic use of acyclovir affect the significance of the results? Please show data
  6. Which will be the contribution of viral infection to OM pathogenesis. Usually a productive infection of epithelial cells induces IFN-alpha and a long series of pro-inflammatory events that hardly happen in patients under chemotherapy. On the other side infected epithelial cells actively express anti-apoptotic genes whose products allow the virus to evade apoptosis and the cells dies from damage induced by viral replication

Author Response

  1. The Authors reported that chemotherapy significantly increased the prevalence of HSV-1 detection: in this context it would be interesting to see the results divided by gender.
  • We found that HSV activation after chemotherapy was more evident in male than female. It explains why we often observed significant association of male with OM incidence or severity in the analyses presented in Table 2. The requested analysis results have been added as Figure 1E, and described in the Result section (lines 186-187) as following: The increase in HSV-1 detection after chemotherapy was particularly evident in male (Figure 1E).

  1. The comparison with the control group should not be the subject of statistical analysis since all patients have been treated with antibiotics.
  • We compared only the baseline communities of patients with the control group because we need some idea on how similar or different those two groups are. The readers will want to know the same thing. Instead, we clearly stated the limitation in Discussion of the revised manuscript (lines 331-335) as following: All patients were also under prophylactic ciprofloxacin and used chlorhexidine gargle during the study period. Notably, the baseline bacterial communities of 20 patients were exposed to ciprofloxacin and chlorhexidine for ≤ 4 days, which should be considered in interpretation of the results that compared the baseline bacterial communities of patients with the control communities.

  1. if anti-fungal therapy was given during the course of the study to all patients: how can the results regarding Candida infection be valid?
  • For hematopoietic stem cell transplant recipients, anti-fungal prophylaxis aims at preventing invasive mold or yeast infection. The prophylaxis contributes to reduce the incidence of fungal infection thereby improve transplant outcomes, but it cannot perfectly prevent fungal infections. Sporadic resistance to echinocandin anti-fungal agents is possible (ref 18). This issue has been addressed in Discussion section (lines 329-331) as following: Despite the uniform prophylactic administration of micafungin and the use of nystatin gargle, colonization with Candida spp. was observed in the 27 evaluations of 18 patients, which may be attributed to echinocandin resistance in Candida

  1. The Authors say that the presence of HSV1 is associated with reduced species eveness and Shannon diversity only in the post-chemotherapy community. In addition to comparing indexes of microbial diversity I strongly suggest to search significant enrichment of microbial taxa in HSV1+ patients compared to HSV1- by LEFSE analysis. Data from heatmap are not sufficient.
  • LEfSe analysis was carried out and presented in Figure 4. The data have been described in Result section (lines 280-282) as following: The presence of HSV-1 was associated with several bacterial taxa, and the associated taxa in the baseline and post-chemotherapy communities were different to each other (Figure 4).

  1. The Authors say: Interestingly, the use of acyclovir was associated with the increased incidence of OM (Odds ratio 3.982, 95% CI 319 1.173–13.514, = 0.027), reflecting increased HSV-1 detection among the acyclovir recipients and the therapeutic use of acyclovir. The answer is: how much did the prophylactic use of acyclovir affect the significance of the results? Please show data.
  • The Odds ratio 3.982 was obtained by generalized estimating equations analysis of 79 post-chemotherapy observations. In the revised manuscript, the effect of acyclovir use on the post-chemotherapy HSV-1 positivity and OM development has been presented in Table S3 using a chi-square test. Among the 20 patients who had post-chemotherapy HSV-1 positivity but no acyclovir, 14 experienced OM. Among the 7 patients who had post-chemotherapy HSV-1 positivity and acyclovir, 6 experienced OM, explaining the OR 3.982. However, looking at the data presented in Table S2, the simple interpretation of the results was that the use of acyclovir did not affect either the post-chemotherapy HSV-1 positivity or the incidence of OM. These have been described in Discussion (lines 322-329) as following: Whereas the 2 patients under the prophylactic acyclovir experienced neither HSV-1 activation nor OM throughout the study period, 7 and 6 of 10 patients who received acyclovir therapy for a short period had post-chemotherapy HSV-1 positivity and OM, respectively. Combining the prophylactic and therapeutic purposes, the use of acyclovir did not affect either the post-chemotherapy HSV-1 positivity or the incidence of OM in the current study (Table S3), probably because the number of prophylactic cases was small and the therapeutic acyclovir was administered after detection of HSV-1 in the oral cavity in most cases.

Table S3. The effect of the use of acyclovir on post-chemotherapy HSV-1 positivity and OM development.

Post-CTx HSV-1

p

OM experience

P

Negative

Positive

No

Yes

The use of acyclovir

No

14 (73.7%)

20 (74.1%)

0.618

20 (76.9%)

14 (70.0%)

0.738

Yes

5 (26.3%)

7 (25.9%)

6 (23.1%)

6 (30.0%)

  1. Which will be the contribution of viral infection to OM pathogenesis. Usually a productive infection of epithelial cells induces IFN-alpha and a long series of pro-inflammatory events that hardly happen in patients under chemotherapy. On the other side infected epithelial cells actively express anti-apoptotic genes whose products allow the virus to evade apoptosis and the cells dies from damage induced by viral replication.
  2. As described Introduction, induction of pro-inflammatory cytokines in response to DAMP underlie the OM pathogenesis. Therefore, we don’t agree on the view of the reviewer that HSV-induced pro-inflammatory events cannot occur in the oral keratinocytes of patients under chemotherapy.

The prevention of apoptosis in HSV-1-infected epithelial cells occurs temporarily, i.e. 3 to 6 hours post infection. HSV-1 induces apoptosis in human keratinocytes before the production of IE gene products, ICP27 proteins, that happens at 3 to 6 hours post infection. In the later phase of lytic replication, HSV-1-infected epithelial cells eventually die from the virus-induced cell damage (Aubert et al. Modulation of apoptosis during HSV infection in human cells. Microbes and Infection 2001;3:859-66).

The HSV-1 observed in the current study seems to be mostly asymptomatic viral shedding. The paragraph discussing the potential contribution of HSV-1 to OM pathogenesis has been edited as following (lines 365-374): Only five of 49 observations with HSV-1 detection accompanied viral lesions. Asymptomatic viral shedding of HSV-1 in seropositive subjects without a history of recurrent herpes labialis has been reported, in which HSV-1 DNA was detected in the buccal mucosa, lower lip, and dorsum of the tongue at a similar frequency [25]. In healthy individuals, the amount and duration of asymptomatic viral shedding of HSV-1 were lower than those from the lesion, explaining the lack of either symptoms or lesion [26]. HSV-1 activates TLR2 and TLR9 and induces the production of inflammatory cytokines IL-1 and IL-6 and chemokines CCL3 and CCL4 from keratinocytes [27, 28]. Together with the reactive oxygen species and inflammatory cytokines induced by the cytotoxic agents, the presence of HSV-1 may contribute to the development of OM under chemotherapy

Reviewer 2 Report

  • The authors performed a nice study with potential relevance and thorough statistical evaluation. However, some serious issues need to be addressed first.
  • It is very common to refer to the day of stem cell transplant as day 0. I would suggest to renumber your days to make a comparison to other studies easier.
  • Lines 111-112: Ten patients received acyclovir for 3–15 days during the study period to treat herpes zoster, disseminated zoster, or severe OM. Oral mucositis is per definition not associated with a herpes infection, so I would rephrase it as ‘severe oral ulcerations’.
  • I would suggest to carefully check in what case a ‘fungal infection’ is meant and in what case specifically an ‘infection with Candida species’ is meant. Please rephrase where necessary throughout the manuscript.
  • Materials and Methods. Please describe whether and how a distinction between Candida species is performed.
  • Please define OM and OM-free. Is a score of 1 at both scales defined as OM? That is not clear from the text. I can imagine that the CTCAE a distinction between grades 0-1 and 2-5 makes sense. In Literature, usually a distinction between ulcerative and non-ulcerative OM is made, because in case of an ulceration there is a breach in mucosa and microbiome may significantly change because of that. Scores on the OMAS scale are even a bit more complicated. So please clearly specify how OM and OM-free are defined and seriously consider making the distinction between ulcerative and non-ulcerative OM.
  • What were the controls? Were they age and gender matched? Free of oral disease (active caries, no pockets > 5mm, no smokers, ..)? Please specify.
  • The comparison between oral bacteria and Candida spp in patients that receive antibiotics and antifungal medication and control subjects has no meaning. It cannot be known what the effect of this medication was on the composition of the oral bacteria and Candida spp. So these should be removed.
  • Results. Lines 208-209. DNA copy number is not the same as bacterial load. There are data that make clear that there is no correlation between copy number and bacterial load. So, this must be rephrased, but then the relevance of the data is unclear. Or these data should be removed and all conclusions based on that these results too. I would suggest the latter option.
  • In week 4 only 6 patients could be evaluated, and these patients were the patients that were the worst. The rest of them was ‘good’ enough to go home. That is such a selection, so it is not possible to use these data as post CT. They should be removed from the manuscript.
  • Discussion. First the results section needs to be adjusted. Then the discussion and the conclusion need to be rewritten, based on those changes.
  • I would suggest to look closely in the literature, very recently some papers about this topic were published from other groups.

Author Response

Authors thank reviewer for detailed and constructive comments.

  1. It is very common to refer to the day of stem cell transplant as day 0. I would suggest to renumber your days to make a comparison to other studies easier.
  • Because the aim of this study was to investigate the role of oral bacteriota and HSV-1 in the incidence and severity of chemotherapy-induced OM, sampling and evaluation of OM should be carried out based on the day of initiation of conditioning chemotherapy. As listed in Table S1, the day of stem cell infusion after the initiation of conditional chemotherapy varied. Renumbering days based on the day of stem cell transplant would be confusing to appreciate the chemotherapy induced OM.
  1. Lines 111-112: Ten patients received acyclovir for 3–15 days during the study period to treat herpes zoster, disseminated zoster, or severe OM. Oral mucositis is per definition not associated with a herpes infection, so I would rephrase it as ‘severe oral ulcerations’.
  • It has been changed.
  1. I would suggest to carefully check in what case a ‘fungal infection’ is meant and in what case specifically an ‘infection with Candida species’ is meant. Please rephrase where necessary throughout the manuscript.
  • Micafungin and nystatin gargle were used to prevent fungal infection, including oropharyngeal candidiasis. The phrase ‘fungal infection’ was used only in line 112 to explain the purpose of prophylactic micafungin. The phrase ‘oropharyngeal candidiasis’ was used only in line 124 to explain the purpose of nystatin gargle. The colonization with Candida species evaluated in the current study does not mean infection with Candida.
  1. Please describe whether and how a distinction between Candida species is performed.
  • In the Experimental Sections (lines 131-134), it has been added as follow: The recorded Candida included C. albicans, which forms dark blue colonies, C. tropicalis, C. lusitaniae, and C. kefy, which form pink colonies, and C. dubliniensis, which forms turquoise or white colonies, on the ChromID agar. However, the result was presented as Candida spp.
  1. Please define OM and OM-free. Is a score of 1 at both scales defined as OM? That is not clear from the text. I can imagine that the CTCAE a distinction between grades 0-1 and 2-5 makes sense. In Literature, usually a distinction between ulcerative and non-ulcerative OM is made, because in case of an ulceration there is a breach in mucosa and microbiome may significantly change because of that. Scores on the OMAS scale are even a bit more complicated. So please clearly specify how OM and OM-free are defined and seriously consider making the distinction between ulcerative and non-ulcerative OM.
  • OM was defined as the NCI-CTCAE grade > 0, and this was specified in Result section and the legends for Table 1, Table 2, Figure 1, and Figure 2.
  1. What were the controls? Were they age and gender matched? Free of oral disease (active caries, no pockets > 5mm, no smokers, ..)? Please specify.
  • For the control group, the inclusion criterion was age ≥ 19 years and the exclusion criteria were as follows: (1) smokers; (2) the use of antibiotics or steroid within the last month; (3) any medication for systemic diseases; and (4) subjects with any lesions in the oral mucosa. As presented in Table 1, the controls were not age and gender matched with the patients. Because either the plaque index or decayed, missing, and filled surface (DMFS) scores was not associated with OM in the previous study, those two parameters were not recorded in the newly recruited patients and controls. The “basic oral examination” has been specified as “visual examination for the presences of oral mucosal diseases and any abnormal changes in the oral mucosa”.

  1. The comparison between oral bacteria and Candida spp in patients that receive antibiotics and antifungal medication and control subjects has no meaning. It cannot be known what the effect of this medication was on the composition of the oral bacteria and Candida spp. So these should be removed.
  • Candida colonization was not evaluated in the control subjects, thus was not compared between patients and control. Bacteriota of the control group was compared only with the baseline communities of patients, because how different the baseline communities of patients are from the control communities is useful information. Instead, we made the limitation clear by commenting it in Discussion lines 333-337 as follow: All patients were also under prophylactic ciprofloxacin and used chlorhexidine gargle during the study period. Notably, the baseline bacterial communities of 20 patients were exposed to ciprofloxacin and chlorhexidine for ≤ 4 days, which should be considered in interpretation of the results that compared the baseline bacterial communities of patients with the control communities.

  1. Lines 208-209. DNA copy number is not the same as bacterial load. There are data that make clear that there is no correlation between copy number and bacterial load. So, this must be rephrased, but then the relevance of the data is unclear. Or these data should be removed and all conclusions based on that these results too. I would suggest the latter option.
  • Because the bacteriota analysis by sequencing provide only information on the relative of amounts of each taxa, we used DNA copy number as the crude estimation of total bacterial loads. We wanted to know if the use of antibiotics reduced bacterial load or OM development is associated with increased bacterial loads but found no differences. As recommended, the data have been deleted.

  1. In week 4 only 6 patients could be evaluated, and these patients were the patients that were the worst. The rest of them was ‘good’ enough to go home. That is such a selection, so it is not possible to use these data as post CT. They should be removed from the manuscript.
  • As described in lines 120-121, hospital discharge of evaluated patients were decided according to the recovery of blood cell counts but not to their general medical condition. The 6 patients stayed hospital more than 4 weeks because they were still dependent on platelet transfusion at that time but none of them underwent serious infectious or other medical condition that postponed hospital discharge. It is also noteworthy that they received stem cell infusion on day 8.
  1. First the results section needs to be adjusted. Then the discussion and the conclusion need to be rewritten, based on those changes.
  • Results and Discussion regarding bacterial load based on DNA copy number have been deleted. 
  1. I would suggest to look closely in the literature, very recently some papers about this topic were published from other groups.
  • The works by Miranda-Silva et al. (2020) and by Laheji et al. (2019) have been cited and discussed: lines 329-331, lines 352-354, lines 364-366, lin

Reviewer 3 Report

The paper is interesting for the knowledge of oral bacteriota and HSV-1 in patients with mucositis, but I have some questions. It would be very important to give way to the method to examine the relevance and limitations of the results afterwards.

Experimental section

Why did the authors not exclude patients with alcoholic habit or other toxic habits? Especially in the control group

Line 70. Can the authors send the checklist of STROBE guidelines?

Line 83. Why did you mention reference 11?

Line 83-84. Is it necessary to include the plaque index and DMFT (DMFS) if it is not commented in the results?

Line 101. What does a “basic oral examination” mean?

Line 107. What dosage of ciprofloxacin and micafungin were prescribed? What are the recommendations of the Center International Blood and Marrow Transplant Research for gut decontamination?

Line 118. “”During the admission period, patients rinsed ....with clorhexidine twice daily.” What do the authors call the “admission period”?””

               How many days or weeks do patients rinse with chlorhexidine?

               Under normal conditions, what is the effect of chlorhexidine on     the oral Bacteriota?

              Is Chlorhexidine a Mucositis Prevention Treatment?

Line 120. In addition, was 5-10ml of oral Nystatin prescribed (swallowing or not swallowing the rinse) for the prevention of oropharyngeal candidiasis infection? How many days? during the study, to determine the presence of candida spp?

Statistical Analysis.

   To perform the statistical analysis, did you use two databases? Why didn't you carry out it all at the R vegan package?

   Previously the statistical analysis, is the meaning of Shannon's index reflected in the methodology?

   What was the scale for evaluating the score of mucositis? (Can be read in the abstract), and for Shannon index?

   Did you consider the 10 patients treated with acyclovir for the treatment of herpes zoster in the statistical analysis?

Results

The patient recruitment period, is the same as that by Hong et al.?

     How many patients come from that study?

     What has been the objective that makes the difference between one and another study. To study oral bacteriota the authors prescribe chlorhexidine rinse and ciprofloxacin? The authors must clarify it.

          What does the present study contribute?

          Are the effects of chlorhexidine as expected?

Author Response

Authors thank reviewer for detailed and constructive comments.

 Experimental section

  1. Why did the authors not exclude patients with alcoholic habit or other toxic habits? Especially in the control group
  • All patients were evaluated and sampled while they were admitted to the hospital. Therefore, they cannot access toxic habits such as alcohol. The control group was recruited within the institution. The participants include the authors of current study, the students or staffs of Seoul National University School of Dentistry. It is not likely that any of the participants had serious toxic habits.
  1. Line 70. Can the authors send the checklist of STROBE guidelines?
  • It has been provided.
  1. Line 83. Why did you mention reference 11?
  • It was mis-cited and changed with Ye et al. 2013 (ref 18 in version 1).
  1. Line 83-84. Is it necessary to include the plaque index and DMFT (DMFS) if it is not commented in the results?
  • It has been deleted as suggested.
  1. Line 101. What does a “basic oral examination” mean?
  • It has been specified as follow (lines 84-86): All of the enrolled patients received a basic dental examination that includes panoramic radiography and visual examination for the presences of a potential infectious focus of dental origin, oral mucosal diseases, and any abnormal changes in the oral mucosa.
  • The basic oral examination that the control subjects received was also specified as follow (lines 104-105): They received visual examination for the presences of oral mucosal diseases and any abnormal changes in the oral mucosa
  1. Line 107. What dosage of ciprofloxacin and micafungin were prescribed? What are the recommendations of the Center International Blood and Marrow Transplant Research for gut decontamination?
  • We used ciprofloxacin 500 mg orally twice a day and micafungin 50 mg via IV infusion once a day, which have been added to the revised manuscript line 111. According to the Guideline cosponsored by the CIBMTR and other major stem cell transplantation research societies, prophylactic use of ciprofloxacin is level B2 recommendation and micafungin is level B1 recommendation, respectively (ref 14). Strength of recommendation category 'B' means 'Should generally be offered'.
  1. Line 118. “”During the admission period, patients rinsed ....with clorhexidine twice daily.” What do the authors call the “admission period”?””
  • The phrase has been edited as following (line 122): “During hospitalization, i.e. the entire study period”.
  1. How many days or weeks do patients rinse with chlorhexidine?
  • As described in line 122, patients used chlorhexidine during hospitalization, and the hospitalization period varied. The criteria for discharge is described in lines 120-121.
  1. Under normal conditions, what is the effect of chlorhexidine on the oral Bacteriota?
  • Under normal condition, chlorhexidine is usually prescribed to control gingivitis, periodontitis, or pericoronitis. However, there is no study that investigated the effect of chlorhexidine on the oral bacteriota, so it is not clear. As an antiseptic antibacterial agent, similar effects with antibiotics are expected. The effect of chlorhexidine has been discussed in lines 354-355 as follow: The role of chlorhexidine cannot be excluded, although its effect on the structure of oral microbiota remains to be clarified.
  1. Is Chlorhexidine a Mucositis Prevention Treatment?
  • The use of chlorhexidine gargle was the part of the institutional protocols to prevent infection of gum origin.
  1. Line 120. In addition, was 5-10ml of oral Nystatin prescribed (swallowing or not swallowing the rinse) for the prevention of oropharyngeal candidiasis infection? How many days? during the study, to determine the presence of candida spp?
  • To clarify the usage method and purpose, the sentence has been edited as following (lines 122-125): During hospitalization, i.e., the entire study period, patients rinsed their oral cavity with normal saline 4 times daily, with chlorhexidine twice daily, and with 5–10 mL of nystatin oral suspension 3 times daily immediately after a meal to reduce the risk of oral infection, including oropharyngeal candidiasis.
  1. To perform the statistical analysis, did you use two databases? Why didn't you carry out it all at the R vegan package?
  • We are more used to SPSS. R was used only for the analyses that can’t be carried out with SPSS.
  1. Previously the statistical analysis, is the meaning of Shannon's index reflected in the methodology?
  • The meaning of Shannon index has been added to Result section (lines 224-225) as follow: Chemotherapy did not affect the total bacterial loads but significantly decreased the Shannon diversity, an index that accounts for both richness and evenness of the species present.
  1. What was the scale for evaluating the score of mucositis? (Can be read in the abstract), and for Shannon index?
  • As described in lines 94-95, OM was graded using two methods, the NCI-CTCAE that ranges from 0 to 3 and the Oral Mucositis Assessment Scale (OMAS) that ranges from 0 to 5. The scale mentioned in Abstract is recorded by the Oral Mucositis Assessment Scale (OMAS). To make it clear, it has been capitalized.
  • It is not clear to us what the reviewer asked regarding the scale for Shannon index. The Shannon index can be any number > 0.
  1. Did you consider the 10 patients treated with acyclovir for the treatment of herpes zoster in the statistical analysis?
  • It was discussed in the original version but has been edited as below (lines 321-331), and supporting data are presented in Table S3.

The variable use of acyclovir during the study period may have had confounding effect on HSV-1 detection and development of OM. Whereas the 2 patients under the prophylactic acyclovir experienced neither HSV-1 activation nor OM throughout the study period, 7 and 6 of 10 patients who received acyclovir therapy for a short period had post-chemotherapy HSV-1 positivity and OM, respectively. Combining the prophylactic and therapeutic purposes, the use of acyclovir did not affect either the post-chemotherapy HSV-1 positivity or the incidence of OM in the current study (Table S3), probably because the number of prophylactic cases was small and the therapeutic acyclovir was administered after detection of HSV-1 in the oral cavity in most cases. Importantly, HSV-1 was detected in less than 5% of samples and was not associated with OM in a study where all patients were subjected to prophylaxis with acyclovir or valacyclovir during allogeneic HSCT procedure [18].

Results

  1. The patient recruitment period, is the same as that by Hong et al.?
  • We realized that we copied the date from the paper by Hong et al. Thanks for pointing it out. The patient recruitment period has been corrected in line 177: From July 2016 to June 2018
  1. How many patients come from that study?
  • Twenty-two patients. It is described in Experimental section lines 79-80.
  1. What has been the objective that makes the difference between one and another study.
  • The previous study investigated the evaluation of the relationship between oral microbial factors (HSV reactivation, Candida colonization, plaque index, and dental caries) and CIOM development in patients with hematological malignancies undergoing intensive chemotherapy or hematopoietic stem cell transplantation (HSCT). As described in Introduction, we found a strong association between HSV-1 and OM in the previous study, but also a possibility that additional factors are involved. The previous study included patients who received induction chemotherapy, consolidative chemotherapy, reinduction or salvage, autologous HSCT, and allogeneic HSCT. Thus, clinical protocols, including the use of antibiotics and acyclovir, were too diverse. For example, antibiotics are prophylactically administered to all HSCT recipients but prescribed whenever patients have fever in other chemotherapy. Acyclovir is prophylactically given to allogeneic HSCT recipients but empirically to other chemotherapy recipients when they present the sign of herpes viral infection. To reduce the number of variables that can affect the prevalence of OM or the oral bacteriota, we decided to narrow the subjects to patients undergoing autologous HSCT, which was chosen based on the prevalence of OM and the size of cases. To explain the difference from the previous study, the sentence describing the aim of study in Introduction (lines 63-66) has been edited as following: In this study, we aimed to investigate the role of oral bacteriota and HSV-1 in the incidence and severity of OM. To reduce the number of variables that can affect oral bacteriota, the current study restricted subjects to patients undergoing autologous HSCT.
  1. To study oral bacteriota the authors prescribe chlorhexidine rinse and ciprofloxacin? The authors must clarify it.
  • Patients with hematological malignancies, particularly those who receive HSCT, have profound immunosuppression. Therefore, the control of infection is more important than anything, and changing the infection control protocols for the study was unethical. In previous studies involving allogeneic HSCT (ref 9) or autologous HSCT (ref 23), patients also received prophylactic ciprofloxacin. Because all patients uniformly received ciprofloxacin and chlorhexidine rinse, the association between the reduced diversity of oral bacteriota and OM severity observed in the current study is valid. The reason for the prophylactic ciprofloxacin has been re-emphasized in Discussion, and we also pointed out that the baseline bacterial communities of 20 patients were exposed to ciprofloxacin and chlorhexidine for ≤ 4 days and this should be considered in interpretation of the results that compared the baseline bacterial communities of patients with the control communities.

Lines 320-321: Because HSCT involves profound immunosuppression, all measures for infection prevention and control were maintained throughout the study.

Lines 333-337: All patients were also under prophylactic ciprofloxacin and used chlorhexidine gargle during the study period. Notably, the baseline bacterial communities of 20 patients were exposed to ciprofloxacin and chlorhexidine for ≤ 4 days, which should be considered in interpretation of the results that compared the baseline bacterial communities of patients with the control communities.

  1. What does the present study contribute?
  • The present study demonstrates that chemotherapy-accompanied decrease in the diversity of oral mucosal bacteriota is associated with the severity of OM in patients undergoing autologous HSCT. The reduced diversity may be caused by prophylactic antibiotics. Although the use of prophylactic antibiotics during the neutropenic period in HSCT has been a standard protocol for decades, reevaluation of this practice has been recently proposed due to rising rates of antibiotic resistance and adverse effects of perturbed microbiome. The adverse effects of perturbed oral microbiome may include aggravation of OM. When the use of antibiotics can’t be avoided, the way to restore oral microbial diversity should be developed. The contribution of the present study has been summarized and emphasized in the last paragraph of Discussion (lines 399-411).

In conclusion, the chemotherapy-accompanied activation of HSV and decrease in the bacterial diversity were associated with the incidence and severity of OM, respectively. A clinical trial to evaluate the effect of prophylactic anti-herpes drugs on the prevention of OM in patients undergoing autologous HSCT is under way. Although the use of prophylactic antibiotics during the neutropenic period in HSCT has been a standard protocol for decades, reevaluation of this practice has been recently proposed due to rising rates of antibiotic resistance and adverse effects of perturbed microbiome [31]. The adverse effects of perturbed oral microbiome may include aggravation of OM. Likewise, the use of chlorhexidine gargle for oral care also needs to be reevaluated. In addition, restoration of microbial diversity using salivary microbiota may be a novel therapeutic option to prevent severe OM.

  1. Are the effects of chlorhexidine as expected?
  • In the current study, the effect of chlorhexidine was not investigated. We suspect that chlorhexidine may contribute to the reduced diversity of oral bacteriota. As we mentioned in line 406, the use of chlorhexidine gargle for oral care needs to be reevaluated.

Round 2

Reviewer 1 Report

I believe that the authors addressed all issues in the revised manuscript.

Author Response

We appreciate the recognition of our efforts.

Reviewer 3 Report

The authors should include the term " limitations at study" before conclusions, such as the use of antibiotics (ciprofloxacin), chlorhexidine, micafungin and nystatin, since it is a study of oral microbiota study.

Author Response

A paragraph listing the limitations of study has been added as follow (lines 398-403): This study has a number of limitations. First, ciprofloxacin, chlorhexidine, micafungin, and nystatin were uniformly used, and their role in the chemotherapy-accompanied dysbiosis of buccal mucosal bacteriota cannot be clarified. Second, a relatively small sample size limited the statistical power when the subjects were subdivided by the presence of HSV-1 and OM experience. Third, one of our original aims to find an additional factor other than HSV-1 involved in the incidence of OM was not achieved.